# LoCo: Local Contrastive Representation Learning

**Yuwen Xiong**
Uber ATG
University of Toronto
yuwen@cs.toronto.edu

**Mengye Ren**
Uber ATG
University of Toronto
mren@cs.toronto.edu

**Raquel Urtasun**
Uber ATG
University of Toronto
urtasun@cs.toronto.edu

## Abstract

Deep neural nets typically perform end-to-end backpropagation to learn the weights, a procedure that creates synchronization constraints in the weight update step across layers and is not biologically plausible. Recent advances in unsupervised contrastive representation learning invite the question of whether a learning algorithm can also be made local, that is, the updates of lower layers do not directly depend on the computation of upper layers. While Greedy InfoMax [39] separately learns each block with a local objective, we found that it consistently hurts readout accuracy in state-of-the-art unsupervised contrastive learning algorithms, possibly due to the greedy objective as well as gradient isolation. In this work, we discover that by overlapping local blocks stacking on top of each other, we effectively increase the decoder depth and allow upper blocks to implicitly send feedbacks to lower blocks. This simple design closes the performance gap between local learning and end-to-end contrastive learning algorithms for the first time. Aside from standard ImageNet experiments, we also show results on complex downstream tasks such as object detection and instance segmentation directly using readout features.

## 1 Introduction

Most deep learning algorithms nowadays are trained using backpropagation in an end-to-end fashion: training losses are computed at the top layer and weight updates are computed based on the gradient that flows from the very top. Such an algorithm requires lower layers to "wait" for upper layers, a synchronization constraint that seems very unnatural in truly parallel distributed processing. Indeed, there are evidences that weight synapse updates in the human brain are achieved through local learning, without waiting for neurons in other parts of the brain to finish their jobs [8, 6]. In addition to biological plausibility aims, local learning algorithms can also significantly reduce memory footprint during training, as they do not require saving the intermediate activations after each local module finish its calculation. With these synchronization constraints removed, one can further enable model parallelism in many deep network architectures [45] for faster parallel training and inference.

One main objection against local learning algorithms has always been the need for supervision from the top layer. This belief has recently been challenged by the success of numerous self-supervised contrastive learning algorithms [54, 22, 44, 11], some of which can achieve matching performance compared to supervised counterparts, meanwhile using zero class labels during the representation learning phase. Indeed, Löwe et al. [39] show that they can separately learn each block of layers using local contrastive learning by putting gradient stoppers in between blocks. While the authors show matching or even sometimes superior performance using local algorithms, we found that their gradient isolation blocks still result in degradation in accuracy in state-of-the-art self-supervised learning frameworks, such as SimCLR [11]. We hypothesize that, due to gradient isolation, lower layers are unaware of the existence of upper layers, and thus failing to deliver the full capacity of a deep network when evaluating on large scale datasets such as ImageNet [16].

---

Work done at Uber ATG

To bridge the gradient isolation blocks and allow upper layers to influence lower layers while maintaining localism, we propose to group two blocks into one local unit and share the middle block simultaneously by two units. As shown in the right part of Fig. 1. Thus, the middle blocks will receive gradients from both the lower portion and the upper portion, acting like a gradient "bridge." We found that such a simple scheme significantly bridges the performance gap between Greedy InfoMax [39] and the original end-to-end algorithm [11].

On ImageNet unsupervised representation learning benchmark, we evaluate our new local learning algorithm, named LoCo, on both ResNet [25] and ShuffleNet [40] architectures and found the conclusion to be the same. Aside from ImageNet object classification, we further validate the generalizability of locally learned features on other downstream tasks such as object detection and semantic segmentation, by only training the readout headers. On all benchmarks, our local learning algorithm once again closely matches the more costly end-to-end trained models.

We first review related literature in local learning rules and unsupervised representation learning in Section 2, and further elaborate the background and the two main baselines SimCLR [11] and Greedy InfoMax [39] in Section 3.2. Section 4 describes our LoCo algorithm in detail. Finally, in Section 5, we present ImageNet-1K [16] results, followed by instance segmentation results on MS-COCO [37] and Cityscapes [15].

## 2  Related Work

**Neural network local learning rules:**  Early neural networks literature, inspired by biological neural networks, makes use of local associative learning rules, where the change in synapse weights only depends on the pre- and post-activations. One classic example is the Hebbian rule [27], which strengthens the connection whenever two neurons fire together. As this can result in numerical instability, various modifications were also proposed [49, 7]. These classic learning rules can be empirically observed through long-term potentiation (LTP) and long term depression (LTD) events during spike-timing-dependent plasticity (STDP) [1, 8], and various computational learning models have also been proposed [6]. Local learning rules are also seen in learning algorithms such as restricted Boltzmann machines (RBM) [53, 28, 29], greedy layer-wise training [5, 3] and TargetProp [4]. More recently, it is also shown to be possible to use a network to predict the weight changes of another network [32, 41, 57], as well as to learn the meta-parameters of a plasticity rule [43, 42]. Direct feedback alignment [46] on the other hand proposed to directly learn the weights from the loss to each layer by using a random backward layer. Despite numerous attempts at bringing biological plausibility to deep neural networks, the performances of these learning algorithms are still far behind state-of-the-art networks that are trained via end-to-end backpropagation on large scale datasets. A major difference from prior literature is that, both GIM [39] and our LoCo use an entire downsampling stage as a unit of local computation, instead of a single convolutional layer. In fact, different downsampling stages have been found to have rough correspondence with the primate visual cortex [51, 60], and therefore they can probably be viewed as better modeling tools for local learning. Nevertheless, we do not claim to have solved the local learning problem on a more granular level.

**Unsupervised & self-supervised representation learning:**  Since the success of AlexNet [35], tremendous progress has been made in terms of learning representations without class label supervision. One of such examples is self-supervised training objectives [20], such as predicting context [17, 47], predicting rotation [18], colorization [59] and counting [48]. Representations learned from these tasks can be further decoded into class labels by just training a linear layer. Aside from predicting parts of input data, clustering objectives are also considered [61, 9]. Unsupervised contrastive learning has recently emerged as a promising direction for representation learning [55, 54, 22, 44, 11], achieving state-of-the-art performance on ImageNet, closing the gap between supervised training and unsupervised training with wider networks [11]. Building on top of the InfoMax contrastive learning rule [55], Greedy InfoMax (GIM) [39] proposed to learn each local stage with gradient blocks in the middle, effectively removing the backward dependency. This is similar to block-wise greedy training [3] but in an unsupervised fashion.

**Memory saving and model parallel computation:**  By removing the data dependency in the backward pass, our method can perform model parallel learning, and activations do not need to be stored all the time to wait from the top layer. GPU memory can be saved by recomputing the

activations at the cost of longer training time [12, 21, 19], whereas local learning algorithms do not have such trade-off. Most parallel trainings of deep neural networks are achieved by using data parallel training, with each GPU taking a portion of the input examples and then the gradients are averaged. Although in the past model parallelism has also been used to vertically split the network [35, 34], it soon went out of favor since the forward pass needs to be synchronized. Data parallel training, on the other hand, can reach generalization bottleneck with an extremely large batch size [52]. Recently, [31, 45] proposed to make a pipeline design among blocks of neural networks, to allow more forward passes while waiting for the top layers to send gradients back. However, since they use end-to-end backpropagation, they need to save previous activations in a data buffer to avoid numerical errors when computing the gradients. By contrast, our local learning algorithm is a natural fit for model parallelism, without the need for extra activation storage and wait time.

## 3 Background: Unsupervised Contrastive Learning

In this section, we introduce relevant background on unsupervised contrastive learning using the InfoNCE loss [55], as well as Greed InfoMax [39], a local learning algorithm that aims to learn each neural network stage with a greedy objective.

### 3.1 Unsupervised Contrastive Learning & SimCLR

Contrastive learning [55] learns representations from data organized in similar or dissimilar pairs. During learning, an encoder is used to learn meaningful representations and a decoder is used to distinguish the positives from the negatives through the InfoNCE loss function [55],

$$\mathcal{L}_{q,k^+,\{k^-\}} = -\log \frac{\exp(q \cdot k^+/\tau)}{\exp(q \cdot k^+/\tau) + \sum_{k^-} \exp(q \cdot k^-/\tau)}. \tag{1}$$

As shown above, the InfoNCE loss is essentially cross-entropy loss for classification with a temperature scale factor $\tau$, where $q$ and $\{k\}$ are normalized representation vectors from the encoder. The positive pair $(q, k^+)$ needs to be classified among all $(q, k)$ pairs. Note that since the positive samples are defined as augmented version of the same example, this learning objective does not need any class label information. After learning is finished, the decoder part will be discarded and the encoder's outputs will be served as learned representations.

Recently, Chen et al. proposed SimCLR [11], a state-of-the-art framework for contrastive learning of visual representations. It proposes many useful techniques for closing the gap between unsupervised and supervised representation learning. First, the learning benefits from a larger batch size (~2k to 8k) and stronger data augmentation. Second, it uses a non-linear MLP projection head instead of a linear layer as the decoder, making the representation more general as it is further away from the contrastive loss function. With $4\times$ the channel size, it is able to match the performance of a fully supervised ResNet-50. In this paper, we use the SimCLR algorithm as our end-to-end baseline as it is the current state-of-the-art. We believe that our modifications can transfer to other contrastive learning algorithms as well.

### 3.2 Greedy InfoMax

As unsupervised learning has achieved tremendous progress, it is natural to ask whether we can achieve the same from a local learning algorithm. Greedy InfoMax (GIM) [39] proposed to learn representation locally in each stage of the network, shown in the middle part of Fig. 1. It divides the encoder into several stacked modules, each with a contrastive loss at the end. The input is forward-propagated in the usual way, but the gradients do not propagate backward between modules. Instead, each module is trained greedily using a local contrastive loss. This work was proposed prior to SimCLR and achieved comparable results to CPC [55], an earlier work, on a small scale dataset STL-10 [14]. In this paper, we used SimCLR as our main baseline, since it has superior performance on ImageNet, and we apply the changes proposed in GIM on top of SimCLR as our local learning baseline. In our experiments, we find that simply applying GIM on SimCLR results in a significant loss in performance and in the next section we will explain our techniques to bridge the performance gap.

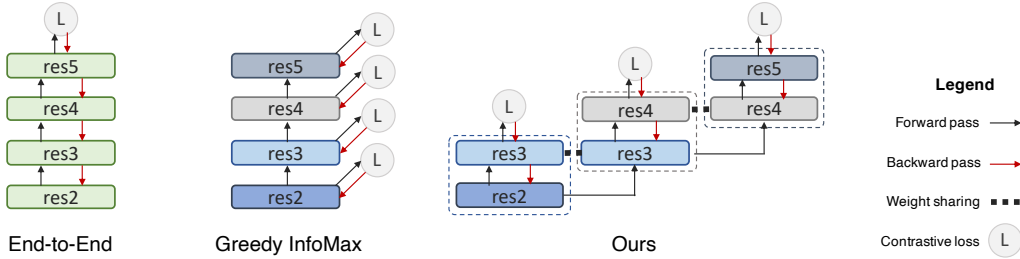

Figure 1: Comparison between End-to-End, Greedy InfoMax (GIM) and LoCo

# 4 LoCo: Local Contrastive Representation Learning

In this section, we will introduce our approach to close the gap between local contrastive learning and state-of-the-art end-to-end learning.

In the left part of Fig. 1, we show a regular end-to-end network using backpropagation, where each rectangle denotes a downsample stage. In ResNet-50, they are *conv1+res2*, *res3*, *res4*, *res5*. In the middle we show GIM [39], where an InfoNCE loss is added at the end of each local stage, and gradients do not flow back from upper stages to lower stages. Our experimental results will show that such practice results in much worse performance on large-scale datasets such as ImageNet. We hypothesize that it may be due to a lack of feedback from upper layers and a lack of depth in terms of the decoders of lower layers, as they are trying to greedily solve the classification problem. Towards fixing these two potential problems, on the right hand side of Fig. 1 we show our design: we group two stages into a unit, and each middle stage is simultaneously shared by two units. Next, we will go into details explaining our reasonings behind these design choices.

## 4.1 Bridging the Gap between Gradient Isolation Blocks

First, in GIM, the feedback from high-level features is absent. When the difficulty of the contrastive learning task increases (e.g., learning on a large-scale dataset such as ImageNet), the quality of intermediate representations from lower layers will largely affect the final performance of upper layers. However, such demand cannot be realized because lower layers are unaware of what kind of representations are required from above.

To overcome this issue, we hypothesize that it is essential to build a "bridge" between a lower stage and its upper stage so that it can receive feedback that would otherwise be lost. As shown in Fig. 1, instead of cutting the encoder into several non-overlapping parts, we can overlap the adjacent local stages. Each stage now essentially performs a "look-ahead" when performing local gradient descent. By chaining these overlapped blocks together, it is now possible to send feedback from the very top.

It is worth noting that, our method does not change the forward pass, even though *res3* and *res4* appear twice in Fig. 1, they receive the same inputs (from *res2* and *res3*, respectively). Therefore the forward pass only needs to be done once in these stages, and only the backward pass is doubled.

## 4.2 Deeper Decoder

Second, we hypothesize that the receptive field of early stages in the encoder might be too small to effectively solve the contrastive learning problem. As the same InfoNCE function is applied to all local learning blocks (both early and late stages), it is difficult for the decoder to use intermediate representation from the early stages to successful classify the positive sample, because of the limitation of their receptive fields. For example, in the first stage, we need to perform a global average pooling on the entire feature map with a spatial dimension of $56 \times 56$ before we send it to the decoder for classification.

In Section 5, we empirically verify our hypothesis by showing that adding convolutional layers into the decoder to enlarge the receptive field is essential for local algorithms. However, this change does not show any difference in the end-to-end version with a single loss, since the receptive field of the final stage is already large enough. Importantly, by having an overlapped stage shared between local units, we effectively make decoders deeper without introducing extra cost in the forward pass, simultaneously solving both issues described in this section.

| Method | Architecture | Acc. | Local |
|---|---|---|---|
| Local Agg. [61] | ResNet-50 | 60.2 | |
| MoCo [22] | ResNet-50 | 60.6 | |
| PIRL [44] | ResNet-50 | 63.6 | |
| CPC v2 [55] | ResNet-50 | 63.8 | |
| SimCLR* [11] | ResNet-50 | 69.3 | |
| SimCLR [11] | ResNet-50 | **69.8** | |
| GIM [39] | ResNet-50 | 64.7 | ✓ |
| LoCo (Ours) | ResNet-50 | 69.5 | ✓ |
| SimCLR [11] | ShuffleNet v2-50 | 69.1 | |
| GIM [39] | ShuffleNet v2-50 | 63.5 | ✓ |
| LoCo (Ours) | ShuffleNet v2-50 | **69.3** | ✓ |

Table 1: ImageNet accuracies of linear classifiers trained on representations learned with different unsupervised methods, SimCLR* is the result from the SimCLR paper with 1000 training epochs.

| Method | Arch | COCO | | Cityscapes | |
|---|---|---|---|---|---|
| | | $AP^{bb}$ | AP | $AP^{bb}$ | AP |
| Supervised | R-50 | 33.9 | 31.3 | 33.2 | 27.1 |
| Backbone weights with 100 Epochs | | | | | |
| SimCLR | R-50 | 32.2 | 29.9 | 33.2 | 28.6 |
| GIM | R-50 | 27.7 (-4.5) | 25.7 (-4.2) | 30.0 (-3.2) | 24.6 (-4.0) |
| Ours | R-50 | 32.6 (+0.4) | 30.1 (+0.2) | 33.2 (+0.0) | 28.4 (-0.2) |
| SimCLR | Sh-50 | 32.5 | 30.1 | 33.3 | 28.0 |
| GIM | Sh-50 | 27.3 (-5.2) | 25.4 (-4.7) | 29.1 (-4.2) | 23.9 (-4.1) |
| Ours | Sh-50 | 31.8 (-0.7) | 29.4 (-0.7) | 33.1 (-0.2) | 27.7 (-0.3) |
| Backbone weights with 800 Epochs | | | | | |
| SimCLR | R-50 | 34.8 | 32.2 | 34.8 | 30.1 |
| GIM | R-50 | 29.3 (-5.5) | 27.0 (-5.2) | 30.7 (-4.1) | 26.0 (-4.1) |
| Ours | R-50 | 34.5 (-0.3) | 32.0 (-0.2) | 34.2 (-0.6) | 29.5 (-0.6) |
| SimCLR | Sh-50 | 33.4 | 30.9 | 33.9 | 28.7 |
| GIM | Sh-50 | 28.9 (-4.5) | 26.9 (-4.0) | 29.6 (-4.3) | 23.9 (-4.8) |
| Ours | Sh-50 | 33.6 (+0.2) | 31.2 (+0.3) | 33.0 (-0.9) | 28.1 (-0.6) |

Table 2: Mask R-CNN results on COCO and Cityscapes. Backbone networks are frozen. "R-50" denotes ResNet-50 and "Sh-50" denotes ShuffleNet v2-50.

## 5   Experiments

In this section, we conduct experiments to test the hypotheses we made in Section 4 and verify our design choices. Following previous works [59, 55, 2, 33, 22], we first evaluate the quality of the learned representation using ImageNet [16], followed by results on MS-COCO [37] and Cityscapes [15]. We use SimCLR [11] and GIM [39] as our main baselines, and consider both ResNet-50 [25] and ShuffleNet v2-50 [40] backbone architectures as the encoder network.

### 5.1   ImageNet-1K

**Implementation details:**   Unless otherwise specified, we train with a batch size of 4096 using the LARS optimizer [58]. We train models 800 epochs to show that LoCo can perform well on very long training schedules and match state-of-the-art performance; we use a learning rate of 4.8 with a cosine decay schedule without restart [38]; linear warm-up is used for the first 10 epochs. Standard data augmentations such as random cropping, random color distortion, and random Gaussian blurring are used. For local learning algorithms (i.e., GIM and LoCo), 2-layer MLPs with global average pooling are used to project the intermediate features into a 128-dim latent space, unless otherwise specified in ablation studies. Following [59, 55, 2, 33, 22], we evaluate the quality of the learned representation by freezing the encoder and training a linear classifier on top of the trained encoders. SGD without momentum is used as the optimizer for 100 training epochs with a learning rate of 30 and decayed by a factor of 10 at epoch 30, 60 and 90, the same procedure done in [22].

**Main results:**   As shown in Table 1, SimCLR achieves favorable results compared to other previous contrastive learning methods. For instance, CPC [55], the contrastive learning algorithm which Greedy InfoMax (GIM) was originally based on, performs much worse. By applying GIM on top of SimCLR, we see a significant drop of 5% on the top 1 accuracy. Our method clearly outperforms GIM by a large margin, and is even slightly better than the end-to-end SimCLR baseline, possibly caused by the fact that better representations are obtained via multiple training losses applied at different local decoders.

### 5.2   Performance on Downstream Tasks

In order to further verify the quality and generalizability of the learned representations, we use the trained encoder from previous section as pre-trained models to perform downstream tasks, We use Mask R-CNN [24] on Cityscapes [15] and COCO [37] to evaluate object detection and instance segmentation performance. Unlike what has been done in MoCo [22], where the whole network is finetuned on downstream task, here we freeze the pretrained backbone network, so that we better distinguish the differences in quality of different unsupervised learning methods.

**Implementation details:**   To mitigate the distribution gap between features from the supervised pre-training model and contrastive learning model, and reuse the same hyperparameters that are selected for the supervised pre-training model [22], we add SyncBN [50] after all newly added layers

| Pretrain Method | COCO-10K | | COCO-1K | |
|---|---|---|---|---|
| | $AP^{bb}$ | AP | $AP^{bb}$ | AP |
| Random Init | 23.5 | 22.0 | 2.5 | 2.5 |
| Supervised | 26.0 | 23.8 | 10.4 | 10.1 |
| Pretrained weights with 100 Epochs | | | | |
| SimCLR | 25.6 | 23.9 | 11.3 | 11.4 |
| GIM | 22.6 (-3.0) | 20.8 (-3.1) | 9.7 (-1.6) | 9.6 (-1.8) |
| Ours | 26.1 (+0.3) | 24.2 (+0.5) | 11.7 (+0.4) | 11.8 (+0.4) |
| Pretrained weights with 800 Epochs | | | | |
| SimCLR | 27.2 | 25.2 | 13.9 | 14.1 |
| GIM | 24.4 (-2.8) | 22.4 (-2.8) | 11.5 (-2.4) | 11.7 (-2.4) |
| Ours | 27.8 (+0.6) | 25.6 (+0.4) | 13.9 (+0.0) | 13.8 (-0.3) |

Table 3: Mask R-CNN results on 10K COCO images and 1K COCO images

in FPN and bbox/mask heads. The two-layer MLP box head is replaced with a *4conv-1fc* box head to better leverage SyncBN [56]. We conduct the downstream task experiments using mmdetection [10]. Following [22], we use the same hyperparameters as the ImageNet supervised counterpart for all experiments, with $1\times$ ($\sim$12 epochs) schedule for COCO and 64 epochs for Cityscapes, respectively. Besides SimCLR and GIM, we provide one more baseline using weights pretrained on ImageNet via supervised learning provided by PyTorch[1] for reference.

**Results:** From the Table 2 we can clearly see that the conclusion is consistent on downstream tasks. Better accuracy on ImageNet linear evaluation also translates to better instance segmentation quality on both COCO and Cityscapes. LoCo not only closes the gap with end-to-end baselines on object classification in the training domain but also on downstream tasks in new domains.

Surprisingly, even though SimCLR and LoCo cannot exactly match "Supervised" on ImageNet, they are 1 – 2 points AP better than "Supervised" on downstream tasks. This shows unsupervised representation learning can learn more generalizable features that are more transferable to new domains.

## 5.3 Downstream Tasks with Limited Labeled Data

With the power of unsupervised representation learning, one can learn a deep model with much less amount of labeled data on downstream tasks. Following [23], we randomly sample 10k and 1k COCO images for training, namely COCO-10K and COCO-1K. These are 10% and 1% of the full COCO train2017 set. We report AP on the official val2017 set. Besides SimCLR and GIM, we also provide two baselines for reference: "Supervised" as mentioned in previous subsection, and "Random Init" does not use any pretrained weight but just uses random initialization for all layers and trains from scratch.

Hyperparameters are kept the same as [23] with multi-scale training except for adjusted learning rate and decay schedules. We train models for 60k iterations (96 epochs) on COCO-10K and 15k iterations (240 epochs) on COCO-1K with a batch size of 16. All models use ResNet-50 as the backbone and are finetuned with SyncBN [50], *conv1* and *res2* are frozen except "Random Init" entry. We make 5 random splits for both COCO-10K/1K and run all entries on these 5 splits and take the average. The results are very stable and the variance is very small ($< 0.2$).

**Results:** Experimental results are shown in Table 3. Random initialization is significantly worse than other models that are pretrained on ImageNet, in agreement with the results reported by [23]. With weights pretrained for 100 epochs, both SimCLR and LoCo get sometimes better performance compared to supervised pre-training, especially toward the regime of limited labels (i.e., COCO-1K). This shows that the unsupervised features are more general as they do not aim to solve the ImageNet classification problem. Again, GIM does not perform well and cannot match the randomly initialized baseline. Since we do not finetune early stages, this suggests that GIM does not learn generalizable features in its early stages. We conclude that our proposed LoCo algorithm is able to learn generalizable features for downstream tasks, and is especially beneficial when limited labeled data are available.

Similar to the previous subsection, we run pretraining longer until 800 epochs, and observe noticeable improvements on both tasks and datasets. This results seem different from the one reported in [13] that longer iterations help improve the ImageNet accuracy but do not improve downstream VOC

| Extra Layers before MLP Decoder | Local | Sharing | Acc. |
|---|---|---|---|
| None | | | 65.7 |
| None | ✓ | | 60.9 |
| 1 conv block | | | 65.6 |
| 1 conv block (w/o ds) | ✓ | | 63.6 |
| 1 conv block | ✓ | | 65.1 |
| 2 conv blocks | ✓ | | 65.8 |
| 1 stage | ✓ | | 65.8 |
| full network | ✓ | | 65.8 |
| 2-layer MLP | | | 67.1 |
| 2-layer MLP | ✓ | | 62.3 |
| Ours | ✓ | ✓ | 66.2 |
| Ours + 2-layer MLP | ✓ | ✓ | **67.5** |

Table 4: ImageNet accuracies of models with different decoder architecture. All entries are trained with 100 epochs.

| Sharing description | Acc. |
|---|---|
| No sharing | 65.1 |
| Upper layer grad only | 65.3 |
| L2 penalty (1e-4) | 65.5 |
| L2 penalty (1e-3) | 66.0 |
| L2 penalty (1e-2) | 65.9 |
| Sharing 1 block | 64.8 |
| Sharing 2 blocks | 65.3 |
| Sharing 1 stage | **66.2** |

Table 5: ImageNet accuracies of models with different sharing strategies. All entries are trained with 100 epochs.

detection performance. Using 800 epoch pretraining, both LoCo and SimCLR can outperform the supervised baseline by 2 points AP on COCO-10K and 4 points AP on COCO-1K.

## 5.4 Influence of the Decoder Depth

In this section, we study the influence of the decoder depth. First, we investigate the effectiveness of the convolutional layers we add in the decoder. The results are shown in Table 4. As we can see from the "1 conv block without local and sharing property" entry in the table, adding one more residual convolution block at the end of the encoder, i.e. the beginning of the decoder, in the original SimCLR does not help. One possible reason is that the receptive field is large enough at the very end of the encoder. However, adding one convolution block with downsampling before the global average pooling operation in the decoder will significantly improve the performance of local contrastive learning. We argue that such a convolution block will enlarge the receptive field as well as the capacity of the local decoders and lead to better representation learning even with gradient isolation. If the added convolution block has no downsampling factor (denoted as "w/o ds"), the improvement is not be as significant.

We also try adding more convolution layers in the decoder, including adding two convolution blocks (denoted as "2 conv blocks"), adding one stage to make the decoder as deep as the next residual stage of the encoder (denoted as "one stage"), as well as adding layers to make each decoder as deep as the full Res-50 encoder (denoted as "full network"). The results of these entries show that adding more convolution layers helps, but the improvement will eventually diminish and these entries achieve the same performance as SimCLR.

Lastly, we show that by adding two more layers in the MLP decoders, i.e. four layers in total, we can observe the same amount of performance boost on all of methods, as shown in the 4th to 6th row of Table 4. However, increasing MLP decoder depth cannot help us bridge the gap between local and end-to-end contrastive learning.

To reduce the overhead we introduce in the decoder, we decide to add one residual convolution block only and keep the MLP depth to 2, as was done the original SimCLR. It is also worth noting that by sharing one stage of the encoder, our method can already closely match SimCLR without deeper decoders, as shown in the third row of Table 4.

## 5.5 Influence of the Sharing Strategy

As we argued in Sec. 4.1 that local contrastive learning may suffer from gradient isolation, it is important to verify this situation and know how to build a feedback mechanism properly. In Table 5, we explore several sharing strategies to show their impact of the performance. All entries are equipped with 1 residual convolution block + 2-layer MLP decoders.

We would like to study what kind of sharing can build implicit feedback. In LoCo the shared stage between two local learning modules is updated by gradients associated with losses from both lower and upper local learning modules. Can implicit feedback be achieved by another way? To answer this question, we try to discard part of the gradients of a block shared in both local and upper local learning modules. Only the gradients calculated from the loss associated with the upper module will be kept to update the weights. This control is denoted as "Upper layer grad only" in Table 5 and the result indicates that although the performance is slightly improved compared to not sharing any encoder blocks, it is worse than taking gradients from both sides.

We also investigate soft sharing, i.e. weights are not directly shared in different local learning modules but are instead softly tied using L2 penalty on the differences. For each layer in the shared stage, e.g., layers in *res3*, the weights are identical in different local learning modules upon initialization, and they will diverge as the training progress goes on. We add an L2 penalty on the difference of the weights in each pair of local learning modules, similar to L2 regularization on weights during neural network training. We try three different coefficients from 1e-2 to 1e-4 to control the strength of soft sharing. The results in Table 5 show that soft sharing also brings improvements but it is slightly worse than hard sharing. Note that with this strategy the forward computation cannot be shared and the computation cost is increased. Thus we believe that soft sharing is not an ideal way to achieve good performance.

Finally, we test whether sharing can be done with fewer residual convolution blocks between local learning modules rather than a whole stage, in other words, we vary the size of the local learning modules to observe any differences. We try to make each module contain only one stage plus a few residual blocks at the beginning of the next stage instead of two entire stages. Therefore, only the blocks at the beginning of stages are shared between different modules. This can be seen as a smooth transition between GIM and LoCo. We try only sharing the first block or first two blocks of each stage, leading to "Sharing 1 block" and "Sharing 2 blocks" entries in Table 5. The results show that sharing fewer blocks of each stage will not improve performance and sharing only 1 block will even hurt.

### 5.6 Memory Saving

Although local learning saves GPU memory, we find that the original ResNet-50 architecture prevents LoCo to further benefit from local learning, since ResNet-50 was designed with balanced computation cost at each stage and memory footprint was not taken into consideration. In ResNet, when performing downsampling operations at the beginning of each stage, the spatial dimension is reduced by $1/4$ but the number of channels only doubles, therefore the memory usage of the lower stage will be twice as much as the upper stage. Such design choice makes *conv1* and *res2* almost occupy 50% of the network memory footprint. When using ResNet-50, the memory saving ratio of GIM is $1.81\times$ compared to the original, where the memory saving ratio is defined as the reciprocal of peak memory usage between two models. LoCo can achieve $1.28\times$ memory saving ratio since it needs to store one extra stage.

We also show that by properly designing the network architecture, we can make training benefit more from local learning. We change the 4-stage ResNet to a 6-stage variant with a more progressive downsampling mechanism. In particular, each stage has 3 residual blocks, leading to a Progressive ResNet-50 (PResNet-50). Table 6 compares memory footprint and computation of each stage for PResNet-56 and ResNet-50 in detail. The number of base channels for each stage are 56, 96, 144, 256, 512, 1024, respectively. After *conv1* and *pool1*, we gradually downsample the feature map resolution from 56x56 to 36x36, 24x24, 16x16, 12x12, 8x8 at each stage with bilinear interpolation instead of strided convolution [25]. Grouped convolution [35] with 2, 16, 128 groups is used in the last three stages respectively to reduce the computation cost. The difference between PResNet-56 and ResNet-50 and block structures are illustrated in the supplementary material.

By simply making this modification without other new techniques [26, 30, 36], we can get a network that matches the ResNet-50 performance with similar computation costs. More importantly, it has balanced memory footprint at each stage. As shown in Table 7, SimCLR using PResNet-50 gets 66.8% accuracy, slightly better compared to the ResNet-50 encoder. Using PResNet-50, our method performs on par with SimCLR while still achieving remarkable memory savings of 2.76 $\times$. By contrast, GIM now has an even larger gap (14 points behind SimCLR) compared to before with ResNet-50, possibly due to the receptive field issue we mentioned in Sec. 4.2.

| Stage | PResNet-50 Mem. (%) | PResNet-50 FLOPS (%) | ResNet-50 Mem. (%) | ResNet-50 FLOPS (%) |
|---|---|---|---|---|
| res2 | 15.46 | 13.50 | 43.64 | 19.39 |
| res3 | 10.96 | 14.63 | 29.09 | 25.09 |
| res4 | 19.48 | 14.77 | 21.82 | 35.80 |
| res5 | 17.31 | 16.62 | 5.45 | 19.73 |
| res6 | 19.48 | 20.45 | - | - |
| res7 | 17.31 | 20.04 | - | - |
| FLOPs | 4.16G | | 4.14G | |

Table 6: Memory footprint and computation percentages for PResNet-50 and ResNet-50 on stage level.

| Method | Acc. | Memory Saving Ratio |
|---|---|---|
| SimCLR | 66.8 | 1× |
| GIM | 52.6 | 4.56× |
| LoCo | 66.6 | 2.76× |

Table 7: ImageNet accuracies and memory saving ratio of Progressive ResNet-50 with balanced memory footprint at each stage. All entries are trained with 100 epochs.

## 6 Conclusion

We have presented LoCo, a local learning algorithm for unsupervised contrastive learning. We show that by introducing implicit gradient feedback between the gradient isolation blocks and properly deepening the decoders, we can largely close the gap between local contrastive learning and state-of-the-art end-to-end contrastive learning frameworks. Experiments on ImageNet and downstream tasks show that LoCo can learn good visual representations for both object recognition and instance segmentation just like end-to-end approaches can. Meanwhile, it can benefit from nice properties of local learning, such as lower peak memory footprint and faster model parallel training.

## Broader Impact

Our work aims to make deep unsupervised representation learning more biologically plausible by removing the reliance on end-to-end backpropagation, a step towards a better understanding of the learning in our brain. This can potentially lead to solutions towards mental and psychological illness. Our algorithm also lowers the GPU memory requirements and can be deployed with model parallel configurations. This can potentially allow deep learning training to run on cheaper and more energy efficient hardware, which would make a positive impact to combat climate change. We acknowledge unknown risks can be brought by the development of AI technology; however, the contribution of this paper has no greater risk than any other generic deep learning paper that studies standard datasets such as ImageNet.

## Funding Disclosure

We do not have any third party funding source to disclose.

## Footnotes

[1] https://download.pytorch.org/models/resnet50-19c8e357.pth

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
