[Supplementary Material]

# Supplementary material of LoCo: Local Contrastive Representation Learning

**Yuwen Xiong**
Uber ATG
University of Toronto
yuwen@uber.com

**Mengye Ren**
Uber ATG
University of Toronto
mren3@uber.com

**Raquel Urtasun**
Uber ATG
University of Toronto
urtasun@uber.com

## 1 Training Curves

We provide training loss curves of SimCLR, GIM and LoCo for a better understanding of the performance gap between them. Contrastive losses computed using outputs from the full ResNet-50 encoder are shown in Fig. 1. For GIM and LoCo, losses from other decoders, including *res2*, *res3*, *res4*, are also provided. As we can see in Fig. 1, the losses of different decoders in LoCo closely match the loss of the decoder in SimCLR during training, with the exception of *res2*, while for GIM this is not the case.

Figure 1: Training loss curves for SimCLR, GIM and LoCo

## 2 Architecture of Progressive ResNet-50

In this section we show the block structure of each stage in Progressive ResNet-50 in Table 1. The block structure of ResNet-50 is also shown here for reference. We downsample the feature map size progressively using bilinear interpolation, and use basic blocks to reduce the memory footprint of earlier stages, and group convolution to reduce the computation cost of later stages to get a model with more balanced computation and memory footprint at each stage. We use this model to show the great potential of LoCo in terms of both memory saving and computation for model parallelism. As it is designed to have 15~20 memory footprint and computation cost per stage, the peak memory usage will be significantly reduced in local learning, and no worker that handles a stage of the encoder will become a computation bottleneck in model parallelism.

| layer | PResNet-50 | | ResNet-50 | |
|---|---|---|---|---|
| | output size | block structure | output size | block structure |
| conv1 | 112×112 | 7x7, 32, stride 2 | 112×112 | 7×7, 64, stride 2 |
| res2_x | 56×56 | 3×3 max pool, stride 2 <br> $\begin{bmatrix} 3\times3,\ 56 \\ 3\times3,\ 56 \end{bmatrix} \times 3$ | 56×56 | 3×3 max pool, stride 2 <br> $\begin{bmatrix} 1\times1,\ 64 \\ 3\times3,\ 64 \\ 1\times1,\ 256 \end{bmatrix} \times 3$ |
| res3_x | 36×36 | $\begin{bmatrix} 3\times3,\ 96 \\ 3\times3,\ 96 \end{bmatrix} \times 3$ | 28×28 | $\begin{bmatrix} 1\times1,\ 128 \\ 3\times3,\ 128 \\ 1\times1,\ 512 \end{bmatrix} \times 4$ |
| res4_x | 24×24 | $\begin{bmatrix} 1\times1,\ 144 \\ 3\times3,\ 144 \\ 1\times1,\ 576 \end{bmatrix} \times 3$ | 14×14 | $\begin{bmatrix} 1\times1,\ 256 \\ 3\times3,\ 256 \\ 1\times1,\ 1024 \end{bmatrix} \times 6$ |
| res5_x | 16×16 | $\begin{bmatrix} 1\times1,\ 256,\ 2\ grps \\ 3\times3,\ 256,\ 2\ grps \\ 1\times1,\ 1024 \end{bmatrix} \times 1$ <br> $\begin{bmatrix} 1\times1,\ 256,\ 2\ grps \\ 3\times3,\ 256,\ 2\ grps \\ 1\times1,\ 1024,\ 2\ grps \end{bmatrix} \times 2$ | 7×7 | $\begin{bmatrix} 1\times1,\ 512 \\ 3\times3,\ 512 \\ 1\times1,\ 2048 \end{bmatrix} \times 3$ |
| res6_x | 12×12 | $\begin{bmatrix} 1\times1,\ 512,\ 16\ grps \\ 3\times3,\ 512,\ 16\ grps \\ 1\times1,\ 2048 \end{bmatrix} \times 1$ <br> $\begin{bmatrix} 1\times1,\ 512,\ 16\ grps \\ 3\times3,\ 512,\ 16\ grps \\ 1\times1,\ 2048,\ 16\ grps \end{bmatrix} \times 2$ | - | - |
| res7_x | 8×8 | $\begin{bmatrix} 1\times1,\ 1024,\ 128\ grps \\ 3\times3,\ 1024,\ 128\ grps \\ 1\times1,\ 4096 \end{bmatrix} \times 1$ <br> $\begin{bmatrix} 1\times1,\ 1024,\ 128\ grps \\ 3\times3,\ 1024,\ 128\ grps \\ 1\times1,\ 4096,\ 128\ grps \end{bmatrix} \times 2$ | - | - |
| | 1×1 | average pool, 1000-d fc | 1×1 | average pool, 1000-d fc |

Table 1: Architectural details of Progressive ResNet-50 and ResNet-50. Output sizes for both models are specified individually

## 3 Representation visualization

In this section we show some visualization results of the learned representation of SimCLR, GIM and LoCo. We subsample images belonging to the first 10 classes of ImageNet-1K from the validation set (500 images in total) and use t-SNE [1] to visualize the 4096-d vector representation from the PResNet-50 encoder. The results are shown in Fig 2. We can see LoCo learns image embedding vectors that can form more compact clusters compared to GIM.

Figure 2: t-SNE visualization results for SimCLR, GIM and LoCo

# 4 Qualitative results for downstream tasks

Last, we show qualitative results of detection and instance segmentation tasks on COCO in Fig. 3.

Figure 3: Qualitative results on COCO-10K, LoCo trained on 800 epochs with ResNet-50 is used to initialize the Mask R-CNN model

# References

[1] L. v. d. Maaten and G. Hinton. Visualizing data using t-sne. *Journal of machine learning research*, 9(Nov):2579–2605, 2008.