[Reviews · NeurIPS 2020]

Review 1

Summary and Contributions: The paper targets at improving local learning, where each neural module (e.g. a stage of ResNet) receive gradients only from a local loss defined on the current module while blocking all the gradients from the upper modules, in the context of contrastive learning. It proposes to allow each module M to receive extra gradients from the local loss of module M+1, to enable the lower layers to implicitly obtain training signals from the upper layers. It also proposes to increase the depth of the decoder to enlarge the receptive field of lower layers which helps their training. Experiments show that local learning is significantly improved via the two proposed methods.

Strengths: 1. The two methods proposed to improve the existing local learning algorithm are both well motivated. They are designed to solve two problems observed by the authors. 2. Empirical results show that local learning is significantly improved.

Weaknesses: The paper claims in the Abstract that "by overlapping local blocks" (i.e. the first proposed method), it "closes the performance gap between local learning and end-to-end contrastive learning algorithms for the first time." However, the presented empirical results can not support the claim. 1. The comparisons with baseline SimCLR in Table-1 are not fair. SimCLR can achieve accuracy of 65.7% without extra layers in the decoder and 67.1% with extra layers according to Table-4. However, Table-1 is comparing SimCLR without extra layers versus the proposed solution with extra layers. 2. The baseline accuracy of SimCLR with ResNet-50 is 65.7% in Table-1 and Table-4, which is NOT consistent with the result from the original SimCLR paper where the accuracy is 69.3% in their Table-6. In this case, the result of the proposed methods is still significantly worse than the baseline. 3. The transfer learning setting in the SimCLR paper is not studied in this paper. In addition, increasing the decoder depth is not a novel contribution.

Correctness: See the above section.

Clarity: Some key expressions in the paper are vague and need improvements. Here are some examples. 1. "we evaluate the quality of the learned representation by training a linear classifier on top of the trained encoders". Do the authors freeze the encoder? Is it the same setting as the "linear evaluation" in the SimCLR paper? If not, why not? 2. "as they do not require saving the intermediate activations". This is not precise. 3. "the classification task is harder by using a larger batch size (~2k to 8k) and stronger data augmentation".

Relation to Prior Work: Basically clear.

Reproducibility: Yes

Additional Feedback: Do the authors try grouping more than two stages into a unit? Will it further improve the results? What's the best practice if memory saving is not the major concern?


Review 2

Summary and Contributions: This submission introduces a way to enhance local contrastive learning by sending implicit feedback from top layers to bottom layers. Specifically, the authors introduce two contributions, one is stacking overlapping stages, the other is deepen the decoder depth. By these two modifications, the proposed LoCo algorithm is able to match the performance of end-to-end contrastive learning method with reduced memory.

Strengths: 1. The proposed stage overlapping method is effective and straightforward. As mentioned in broader impact section, local contrastive learning can reduce memory footprint of model training, and enable parallel training. 2. Ablation study is complete and convincing. 3. The changes to ResNet is interesting.

Weaknesses: 1. In Table 4, the authors mention SimCLR without 2-layer MLP only drops 1.4% accuracy. However, in original SimCLR paper figure 8, they show that without such head reduces the performance by more than 10 percent. Besides, adding 2-layer MLP to MOCO also brings more than 5 percent improvement. So I'm confused here, and has doubts on the correctness of numbers in table 4. Could authors clarify on this? Also, could authors provide more insights on why a 2-layer MLP or a deep decoder helps the representation learning? This will make the paper stronger. 2. In Table 5, no sharing has a performance of 65.1, which is close to the best 66.2. This weakens the claim of sharing features between stages. 3. Why random initialization of a neural network has good performance as shown in Table 3? 4. When performing downstream tasks, authors use SyncBN, which is not adopted in SimCLR or other papers. This may not be a fair comparison.

Correctness: Yes, the method is correct to my best knowledge. I have some doubts on the correctness of experimental results as mentioned in weakness section point 1.

Clarity: Yes, the paper is well written.

Relation to Prior Work: Yes, it is clearly discussed.

Reproducibility: Yes

Additional Feedback: Post-rebuttal: I have read the authors' rebuttal. I think it addressed most of my concerns. I will keep my score as 6.


Review 3

Summary and Contributions: While backpropagation is an effective approach for training deep neural networks, it is unclear how the brain could implement backpropagation as (1) it would require synchronization across the whole network and (2) because of the weight transport problem (feedback weights must be the same as feedforward weights). In this interesting study, the authors modify SimCLR, a contrastive learning algorithm, by replacing backpropagation of the global objective with multiple local contrastive objectives, each acting on one module of the network. In order to allow information to flow from top modules to bottom modules, they ingeniously intertwine the loss functions acting on adjacent modules. This biologically plausible algorithm achieves performance on par with SIMCLR on ImageNet classification and various downstream tasks (COCO and Cityscapes).

Strengths: - The main claims are sound and backed up by solid empirical evidence. - The intertwining method proposed is novel to my knowledge and is an ingenious solution to bridge the performance gap between Greedy InfoMax methods and backpropagation. - The combination of a SOTA model of self-supervised learning with a biologically-plausible local learning rule is very appealing to a neuroscience audience because it adresses two of the most biologically implausible aspects of traditional deep learning in a single model: the reliance on backpropagation and the reliance on extensively labeled data.

Weaknesses: Biological Plausibility: Although the method avoids backpropagation throughout the whole network, it still seems to rely on backpropagation within the layers constituting a module. Do the authors see this more local version of backpropagation as biologically plausible? It would be important to see this point discussed. Computational Efficiency: a main advantage of Greedy Infomax is that the modules can be trained sequentially, since the fate of the bottom modules do not depend on the top modules. With the intertwining method proposed, this sequential training is no longer possible, so the main advantage of Greedy Infomax seems lost. The authors do show that their method achieves modest memory savings compared to backprop (2.76× in the best case scenario), but it is unclear to me how significant this result is.

Correctness: The main claims of the paper are sound and the methods are correct. These minor claims are less clear to me: Abstract: "we are also the first to show results on complex downstream tasks such as object detection and instance segmentation directly using readout features." => In what sense is this study "the first" to show that? Conclusion: "Meanwhile, it can benefit from nice properties of local learning, such as lower peak memory footprint and less waiting time in model parallelism." => What do the authors mean by "less waiting time"?

Clarity: Yes, the paper is clearly written.

Relation to Prior Work: A discussion of the pros and cons of this method (in terms of biological plausibility and computational efficiency) compared to Feedback Alignement would be very interesting. (e.g. https://www.nature.com/articles/ncomms13276)

Reproducibility: Yes

Additional Feedback: EDIT: I have read the other reviews and the careful rebuttal and maintain my score at 7 (a good submission; accept). I am confident that this work is a valuable contribution to the question of biologically plausible alternatives to backpropagation. Minor: I did not understand this sentence: "Importantly, by having an overlapped stage 173 shared between local units, we effectively make decoders deeper without introducing extra cost in the 174 forward pass, simultaneously solving both issues described in this section." typo line 252: "be"


Review 4

Summary and Contributions: This paper proposes a method to learn representations based on the local contrastive representation learning method. Previously, local representation learning was worse than contrastive learning methods. Hence, the authors tried to close the performance gap between local learning and end-to-end contrastive learning algorithms for the first time.

Strengths: It shows more efficient than end-to-end contrastive learning algorithms, and it gives similar performance.

Weaknesses: This paper fails to show their advantages. The accuracies are not better than previously proposed methods like the SimCLR method. The authors said the proposed method is more efficient than end-to-end contrastive learning methods, but it is not enough to be accepted to the conference. Also, the proposed method is straight-forward, which is just in the middle between end-to-end methods and local learning methods.

Correctness: Yes.

Clarity: Yes.

Relation to Prior Work: Yes, it is clear.

Reproducibility: Yes

Additional Feedback:

[Author Response · NeurIPS 2020]

We thank the reviewers for their time and insightful comments. We address the specific concerns and questions below.

**R1 R3 Table 1, 4:** Sorry if it was not very clear in the paper. SimCLR with 65.7% in Table 1 is the original SimCLR model with 2-layer MLP in the decoder. In Table 4, we find that by adding an extra 2-layer MLP (4 layers in total) in the decoder can further boost the performance to 67.1%. The 1.4% accuracy gap is comparing 4-layer MLP versus 2-layer MLP, not 2-layer MLP versus no MLP. Therefore our conclusion does not contradict the SimCLR or MoCo papers. Meanwhile, adding two more MLP layers can also help boost performance of GIM: 62.3% (Row 5) vs 60.9% (Row 4) and LoCo: 67.5% (we ran this in addition) vs 66.2%. Since the gain of increasing MLP depth is orthogonal to our paper, we kept the decoder MLP depth as 2 (explained in L264). Note that LoCo contains 1 more conv block in the decoder, but we find that adding conv block into SimCLR does not help (Table 4 Row 3). Hence we believe the comparison is fair.

**R1 Results worse than original SimCLR paper:** Our results in the main paper are trained with 100 epochs for faster experimentation and the performance is not saturated. The results match Fig.9 in the SimCLR paper which shows performance also with 100 epochs. We also include results trained with 800 epochs in the Supp. Material. SimCLR and LoCo are 69.8% and 69.5% respectively, which match the 69.3% results in the original SimCLR paper.

**R1 Transfer learning not studied:** Although we did not include CIFAR or Flowers in the paper, we use the more challenging object detection and instance segmentation tasks on COCO and Cityscapes as our transfer learning setting.

**R1 Clarity:** 1. "we evaluate ...": As stated "Following [56, 52, 2, 32, 21]", the encoder is frozen. 2. "'as they do not ...": Thanks for pointing this out, we intend to say that the intermediate activations can be released when the local module finish its calculation. 3. "'the classification task is ...": Sorry for the vague expression. We were trying to describe the contribution made by SimCLR. We will revise these sentences.

**R1 increasing decoder depth:** We believe properly increasing the decoder depth is non-trivial, as adding conv blocks into SimCLR decoder will not help, and simply increasing MLP depth does not help reduce the gap between end-to-end and local learning. We think this is a meaningful finding.

**R3 Benefits of sharing features between stages:** Our goal is to bridge the gap from two different aspects, sharing stage and making decoder deeper by adding convolution. In fact, by only sharing, we can achieve 4 points gain and achieve 64.9% compared to GIM 60.9%. We will include this in the next version. Only adding convolution into the decoder can get 65.2%. The performance eventually achieves 66.2% by combining both improvements.

**R3 Performance of random init:** Although the network is randomly initialized, the whole network (including the backbone) is trained with supervised learning. Similar observations can be found in [3].

**R3 SyncBN:** During the contrastive learning phase, all models including SimCLR and GIM are equipped with SyncBN in our implementation. For downstream tasks, all models use the same head (L208-210), thus SimCLR and GIM also use SyncBN. When finetuning, all models, including the random initialized one, use SyncBN (L227-228).

**R4 Biologically plausibility:** We thank the reviewer for bringing up this interesting point and we will add more discussion to our paper. Although our algorithm still relies on backprop within a module that consists of multiple layers, a layer in a CNN may not strictly map to a "layer" in the brain. In fact, a few residual blocks grouped together are found to roughly correspond to different regions in the visual cortex (e.g. V1, V2, IT) [1]. Local forward pass and backward pass could be seen as rough approximation of what is being done in the brain using recurrent computation [2]. But how to make it more biologically plausible on a local level is a very exciting future direction!

**R4 Computational efficiency:** We have not experimented with sequential greedy training and it would be an interesting future step. However, as hypothesized in the paper, one potential drawback of sequential training in GIM is that the lower unit becomes "unaware" of the computation of the upper unit and may not provide the most useful representation. $2.76\times$ memory saving is still substantial. It can make a sizeable difference in memory intensive tasks such as semantic segmentation, where we can hardly fit a single example per GPU.

**R4 Clarity:** 1. "we are also ...": we believe performing transfer learning on complex downstream tasks like instance segmentation is essential and valuable to evaluate the quality and transferability of the features learned by contrastive learning. We use the word "the first" as there is no other result that can be referred when we submitted the paper. 2. "Meanwhile, it can ...": For end-to-end learning, lower layers are required to wait for the gradients from upper layers, and they might be computed in another machine in model parallelism. In local learning, the dependency can be removed and the waiting time can be less. 3. "Importantly, by having ...": The shared overlapped stage can be treated as part of the decoder for the lower local unit. We believe in this way we effectively make decoders deeper without introducing extra cost in the forward pass (as the computation can be shared for the upper local unit). We will revise these sentences.

**R4 Prior work:** Thanks for pointing it out, we will compare with Feedback Alignment in the next version.

**R5 Advantages of this paper:** The goal of this paper is to propose a simple and effective local learning algorithm that can perform as good as SimCLR. We believe the properties of local learning, including lower peak memory footprint and biological plausibility, are very relevant and interesting to the NeurIPS community.

[1] C. Zhuang, S. Yan, A. Nayebi, M. Schrimpf, M. C. Frank, J. J. DiCarlo, and D. L. K. Yamins. Unsupervised neural network models of the ventral visual stream. *bioRxiv, 10.1101/2020.06.16.155556*, 2020

[2] Q. Liao, and T. Poggio. Bridging the Gaps Between Residual Learning, Recurrent Neural Networks and Visual Cortex. *arXiv preprint 1604.03640*, 2016.

[3] K. He, R. Girshick, and P. Dollár. Rethinking ImageNet Pre-training. *arXiv preprint 1811.08883*, 2018.


[Meta-Review · NeurIPS 2020]

Reviewers were satisfied by the author's response and clarifications. Discussion phase also contributed to harmonizing their view on the relevance and usefulness of well-working local criteria. As a result, R1 and R5 increased their score. The consensus is that the work is a novel and valuable contribution to research on local un/self-supervised learning criteria, with potential relevance for memory savings and biologically plausible alternatives to backpropagation. The AC agrees and recommends acceptance.